# Intra-Individual Comparison of ^124^I-PET/CT and ^124^I-PET/MR Hybrid Imaging of Patients with Resected Differentiated Thyroid Carcinoma: Aspects of Attenuation Correction

**DOI:** 10.3390/cancers14133040

**Published:** 2022-06-21

**Authors:** Hong Grafe, Maike E. Lindemann, Manuel Weber, Julian Kirchner, Ina Binse, Lale Umutlu, Ken Herrmann, Harald H. Quick

**Affiliations:** 1Department of Nuclear Medicine, University Hospital Essen, University Duisburg-Essen, Hufelandstr. 55, 45147 Essen, Germany; manuel.weber@uk-essen.de (M.W.); sekretariat.nuklearmedizin@uk-essen.de (I.B.); ken.herrmann@uk-essen.de (K.H.); 2High-Field and Hybrid MR Imaging, University Hospital Essen, University Duisburg-Essen, 45147 Essen, Germany; maike.lindemann@uk-essen.de (M.E.L.); harald.quick@uk-essen.de (H.H.Q.); 3Department of Diagnostic and Interventional Radiology, Medical Faculty, University Hospital Dusseldorf, 40225 Dusseldorf, Germany; julian.kirchner@med.uni-duesseldorf.de; 4Department of Diagnostic and Interventional Radiology and Neuroradiology, University Hospital Essen, University Duisburg-Essen, 45147 Essen, Germany; lale.umutlu@uk-essen.de; 5Erwin L. Hahn Institute for Magnetic Resonance Imaging, University Duisburg-Essen, 45141 Essen, Germany

**Keywords:** ^124^I-PET/CT, ^124^I-PET/MR, differentiated thyroid carcinoma, attenuation correction (AC), CT-based AC, MR-based AC, quantitative PET imaging

## Abstract

**Simple Summary:**

This study evaluates the qualitative and quantitative differences between 124-iodine PET/CT and PET/MR in oncologic patients with differentiated thyroid carcinoma after thyroidectomy. The impact of improved MR-based attenuation correction (AC) using a bone atlas was analysed in PET/MR data. Despite different patient positioning and AC methods PET/CT and PET/MR provide overall comparable results in a clinical setting. The overall number of detected ^124^I-active lesions and the measured average SUV_mean_ values for congruent lesions were higher for PET/MR when compared to PET/CT. The addition of bone to the MR-based AC in PET/MR slightly increased the SUV_mean_ values for all detected lesions.

**Abstract:**

Background: This study evaluates the quantitative differences between 124-iodine (I) positron emission tomography/computed tomography (PET/CT) and PET/magnetic resonance imaging (PET/MR) in patients with resected differentiated thyroid carcinoma (DTC). Methods: *N* = 43 ^124^I PET/CT and PET/MR exams were included. CT-based attenuation correction (AC) in PET/CT and MR-based AC in PET/MR with bone atlas were compared concerning bone AC in the head-neck region. AC-map artifacts (e.g., dentures) were noted. Standardized uptake values (SUV) were measured in lesions in each PET data reconstruction. Relative differences in SUV_mean_ were calculated between PET/CT and PET/MR with bone atlas. Results: Overall, *n* = 111 ^124^I-avid lesions were detected in all PET/CT, while *n* = 132 lesions were detected in PET/MR. The median in SUV_mean_ for *n* = 98 congruent lesions measured in PET/CT was 12.3. In PET/MR, the median in SUV_mean_ was 16.6 with bone in MR-based AC. Conclusions: ^124^I-PET/CT and ^124^I-PET/MR hybrid imaging of patients with DTC after thyroidectomy provides overall comparable quantitative results in a clinical setting despite different patient positioning and AC methods. The overall number of detected ^124^I-avid lesions was higher for PET/MR compared to PET/CT. The measured average SUV_mean_ values for congruent lesions were higher for PET/MR.

## 1. Introduction

Following the introduction of hybrid position emission tomography/computed tomography (PET/CT) in 2000, it has increasingly become the modality of choice for diagnosis and therapy monitoring of various oncologic diseases [1]. In 2010, PET/magnetic resonance (PET/MR) was added as a new hybrid imaging modality for clinical application [2,3,4,5]. PET/MR imaging inherently offers improved soft-tissue contrast and additional functional imaging parameters (e.g., diffusion-weighted imaging, DWI) compared to PET/CT imaging. For the detection and staging of differentiated thyroid carcinoma (DTC) both hybrid imaging modalities, PET/CT and PET/MR, have been used with radiotracers ^124^Iodine-NaI (^124^I) or fluoro-18-fluorodeoxyglucose (^18^F-FDG) [6,7,8,9,10,11]. The high sensitivity of the PET component combined with the superior ability of MR to detect small lesions in the head/neck region may result in improved diagnostics, therapy monitoring and ^124^I dosimetry of thyroid cancer compared with PET/CT [9,10,11]. PET/CT and PET/MR examinations are performed differently with regard to patient positioning and acquisition times and require fundamentally different attenuation correction (AC) methods.

For optimal PET image quality and accurate PET quantification, precise attenuation correction maps (AC maps) of the patient tissues are needed in both, PET/CT and PET/MR. The accuracy and repeatability of such patient-specific AC maps are important preconditions to quantify in vivo biodistribution by virtue of PET. In PET/CT imaging, CT data (given in Hounsfield units, HU) can be converted to the linear attenuation coefficients (LAC) at a PET energy level of 511 keV by a bilinear conversion [12,13]. Thus, CT-based AC results in an AC model of the patient–individual anatomy and continuous LAC values of the patient tissues, also including bone information. The AC of human soft tissues in integrated PET/MR, on the other hand, has to rely on MR imaging providing proton densities and tissue-dependent spin relaxation properties, but not electron densities. Thus, MR images do not contain information about the photon attenuation magnitude and, thus, cannot be converted to LAC at 511 keV as needed for AC of PET data.

The established standard MR-based AC in whole-body PET/MR is a segmentation-based approach [14,15]. Here, for example, a Dixon-VIBE (volumetric interpolate breath-hold examination) MR sequence provides tissue segmentation into four different tissue compartments (background air, lung, fat, soft tissue) with predefined linear attenuation coefficients [15]. The Dixon-VIBE AC map is a discrete AC model (four-compartment AC map) and does not provide attenuation information for highly attenuating bone tissue. In the initial standard MR-based AC methods, bones are assigned to the LAC of soft tissue, which may lead to a systematical underestimation in PET signal [16]. The addition of a dedicated bone atlas into the four-compartment MR-based AC was, for example, introduced by Paulus et al. [17] and today can be considered an established method in MR-AC. The bone atlas adds LAC of bone tissue as a fifth compartment to the Dixon-VIBE AC map (five-compartment AC map) [17,18]. This model-based approach applies continuous LAC (0.1 cm^−1^ up to 0.2485 cm^−1^) of six major bones (skull, spine, pelvis and upper femurs) to the Dixon-VIBE AC map to improve MR-based AC in whole-body PET/MR. The bone model is registered to the individual Dixon-VIBE MR images of each patient. The improved MR-based AC method including the bone model has been evaluated in a clinical setting using ^18^F-FDG whole-body PET/MR imaging and provided improved PET lesion detection and quantification in these studies [19,20].

Against this background, this intra-individual comparison study evaluates the qualitative and quantitative differences between ^124^I-PET/CT and ^124^I-PET/MR imaging in patients with differentiated thyroid carcinoma (DTC) after thyroidectomy. The evaluation was performed retrospectively on hybrid imaging data that were acquired with standard imaging and PET reconstruction parameters specific for PET/CT and PET/MR examinations to identify and quantify existent differences of both hybrid modalities in this specific clinical application. The second aim of this study was to analyze the specific impact of improved MR-based AC using a bone atlas in PET/MR data [17].

## 2. Materials and Methods

### 2.1. Patient Population

In this retrospective single-administration dual hybrid imaging study, patients with DTC after thyroidectomy who underwent ^124^I-PET/CT and ^124^I-PET/MR on clinical indication from June 2016 until January 2021 were included. All patients received a whole-body (skull base to mid-thigh) PET/CT examination and an additional dedicated head-neck (skull base to upper lung) PET/MR examination allowing for an intra-individual comparison of the same anatomical body region. According to the established clinical standard protocol [7,21,22], PET/CT and PET/MR were acquired on the same day approximately 24 h post ^124^I administration (physical half-life: 4.2 days). Initially, *n* = 38 patients with overall *n* = 48 PET/CT and PET/MR head-neck examinations were considered who presented with iodine-positive lesions on both hybrid modalities. Three patients with five examinations were excluded from the study because of obvious errors in the MR-based AC map (segmentation errors in the Dixon-VIBE AC map due to metal implants, and/or registration errors of the bone atlas due to BMI > 40 kg/m^2^) (Figure 1). In total, *n* = 35 patients with *n* = 43 examinations were included in this study. The 23 female and 12 male patients had a mean age of 52 years (range 16–85 years) and the mean BMI was 27 kg/m^2^ (range 18–39 kg/m^2^). The mean ± standard deviation of the administered ^124^I activity was 34.5 ± 9.9 MBq. The mean ± standard deviation of the interval between tracer administration and PET/CT scanning was 24 h and 35 min ± 2 h and 47 min. The average time span from tracer administration to PET/MR scanning was 28 h and 53 min ± 5 h and 7 min. Further information about the patient population is listed in Table 1. Written informed consent was given before PET/CT and PET/MR examinations. All procedures performed were in accordance with the ethical standards of the institutional review board of the Medical Faculty of the University Duisburg-Essen (EC approval number: 11-4822-4825-BO) and with the principles of the Declaration of Helsinki and its later amendments.

### 2.2. Hybrid Image Acquisition

All patient examinations were performed on a whole-body PET/CT system (Biograph mCT, Siemens Healthcare GmbH, Erlangen, Germany) and subsequently on an integrated whole-body 3 Tesla PET/MR system (Biograph mMR, Siemens Healthcare GmbH, Erlangen, Germany).

In PET/CT, the patients were positioned with arms resting above the head. CT measurement for AC was acquired in low-dose technique (without i.v. contrast agent) with a tube voltage of 120 kVp, tube current time product of 15 mAs, beam pitch of 1.0, and 5 mm slice thickness. The whole-body PET/CT emission data were acquired from head to thigh with five to eight bed positions and 4 min PET acquisition for each bed position. The CT data were reconstructed with a voxel size of 0.98 × 0.98 × 1.5 mm^3^ and standard reconstruction of kernel B30f. PET/CT is referred to as the reference standard in this comparison.

In PET/MR, patients were positioned with arms resting beside the body. MR measurement for AC was acquired using a Dixon-VIBE sequence with the following sequence parameters: parallel imaging acceleration factor R = 5, matrix 390 × 240 with 1.3 × 1.3 mm in-plane pixel size, 136 slices each 3.0 mm, flip angle 10°, repetition time (TR) 3.8 ms, echo times (TE) TE1/TE2 1.2/2.4 ms. PET/MR data acquisition was limited to the head/neck region only and was acquired for 20 min for a single bed station. Figure 2 exemplarily shows an intra-individual side-by-side comparison of PET/CT and PET/MR data acquired in a patient with a left-cervical ^124^I-avid lesion (Figure 2).

### 2.3. Attenuation Correction

In PET/CT, the CT-based attenuation data were transformed to LAC of PET at 511 keV using the implemented product version of HU-to-LAC conversion of the PET/CT system resulting in a continuous AC model with patient individual bone anatomy. In PET/MR, the generation of the MR-based AC of the patient tissues is more complex. MR images of the patient tissues acquired with the Dixon-VIBE sequence are used for AC. These MR images are then automatically segmented into four tissue compartments. For each tissue compartment a fixed and predefined LACs is assigned: soft tissue 0.1 cm^−1^, fat 0.0854 cm^−1^, lung 0.0224 cm^−1^ and background air 0.0 cm^−1^. This discrete MR-based AC model providing four tissue compartments is referred to henceforth as the four-compartment AC map [4,5,15]. Additionally, in this study, a second MR-based AC method has been applied to all PET/MR data adding bone tissue as a fifth tissue compartment to the high-resolution CAIPIRINHA Dixon-VIBE sequence. This additional reconstruction applies a model-based bone segmentation algorithm [17,18,23] and adds the major bones (skull, spine, pelvis, and femurs) as a fifth tissue compartment to the previously mentioned four tissue compartments (air, lung, muscle, soft tissue, bone). The model-based bone atlas adds pre-registered bone mask pairs to the resulting MR-based AC with continuous LACs for bone tissue ranging from 0.1 cm^−1^ up to 0.2485 cm^−1^. Detailed information on this bone model is provided by Paulus et al. [17].

### 2.4. Image Reconstruction and Analysis

The PET/CT data were reconstructed using the ordered subsets expectation maximization (OSEM) algorithm with time-of-flight (TOF) with 21 subsets and 2 iterations and a retrospectively reconstructed three-dimensional Gaussian filter of 3 mm and a reconstructed (cuboid-shaped, isotropic) voxel size with a side length of 2.0 mm in each dimension. PET/MR data were retrospectively reconstructed using the image data reconstruction tool provided by the PET/MR system manufacturer (e7 tools, Siemens Molecular Imaging, Knoxville, TN, USA). Two PET reconstructions per patient and examination were generated: (1) MR-based AC without bone (four-compartment AC map) and (2) MR-based AC with bone atlas (five-compartment AC map). The PET data in both reconstructions were reconstructed using ordinary Poisson ordered subsets expectation maximization (OP-OSEM) with 3 iterations and 21 subsets and a 4 mm Gaussian filter resulting in a matrix of 344 × 344 × 127 (resolution 2.09 × 2.09 × 2.03 mm^3^).

To assess and compare the bone volume and LAC values in CT-based and MR-based AC for all 43 examinations, the CT-based AC map was registered to the MR-based AC map and was cut in a longitudinal direction to match the head-neck field-of-view of MR-based AC. In the CT-based AC maps, only the skull and the spine were considered in the analysis (excluding shoulders, upper arms, ribs, sternum, and clavicles) to match the bones available in the MR-based AC with the bone model (Figure 3). Bone tissue was segmented in the CT- and MR-based AC map and the volume of bone tissue (relative to the total volume of the MR-based AC map) was measured as well as the LAC values.

An experienced radiologist and an experienced nuclear medicine specialist in consensus assessed all three PET patient datasets per patient (PET/CT and PET/MR with and without bone). Image reading entailed identifying up to five lesions per patient with focally increased radiotracer uptake. Volumes of interest (VOI) were placed around the detected lesions and the standardized uptake values (SUV_mean_ and SUV_max_) in all PET datasets from PET/CT and PET/MR of each patient and examination were measured. In order to ensure accurate and identical placement of all VOIs in all PET data reconstructions, VOIs were delineated with the help of a *syngo*.via a workstation (Siemens Healthcare GmbH).

Relative differences were calculated to evaluate the quantitative difference between PET data from PET/CT vs. PET data from PET/MR reconstructed two times, with and without bone atlas in the MR-based AC, respectively.

Bland–Altman plots were generated to assess the general quantitative difference between PET/CT and PET/MR, where the PET/MR data were reconstructed twice, with MR-based AC with and without bone information. Descriptive statistics were used to calculate the mean values and standard deviation of all measured SUVs and the relative differences for all detected lesions in all examinations.

## 3. Results

The intra-individual comparison between CT-based and MR-based bone information in the AC maps of all patient data sets resulted in a measured bone volume of 4.48 ± 1.08% for CT-based AC and 3.99 ± 0.96% for MR-based AC. The range of LAC was 0.111–0.266 for CT-based AC and 0.101–0.247 for MR-based AC and the mean values of LAC were 0.134 ± 0.021 for CT-based AC and 0.128 ± 0.002 for MR-based AC (Table 2). The deviation between both AC methods is rather small and the results of bone volume and LAC values are in good agreement between CT-based AC and MR-based AC in the head-neck region. Note that the measured bone volume of CT-based AC and the corresponding LAC are slightly higher than in the MR-based bone atlas. Here, small but highly attenuating metal implants (LAC > 0.250, mainly dental fillings and implants) in the CT-based AC account for this observation, albeit with only neglectable volume in comparison to the total AC-map volume. In MR-based AC, on the other hand, dental fillings and small implants do not result in visible artifacts and thus do not contribute to higher (or lower) LAC in these regions (Table 2).

In Figure 4, two patient examples with typical AC-map artifacts are given. CT-based AC maps often reveal streak and beam hardening artifacts around metal implants (e.g., dental fillings and implants). Such artifacts might result in locally increased LAC in the CT-based AC that may result in a local overestimation in the PET signal. Nearly all patient data in this study revealed metal-based artifacts in the CT-based AC maps around dental implants. Patient example 1 in Figure 4 was thus also included in this study. Larger metal implants may also cause artifacts in the MR-based AC. Here, signal voids in the MR images due to metal may lead to segmentation errors in the Dixon-VIBE MR-based AC, and signal voids due to metal artifacts may then be segmented as air. Soft tissue and the right lung in the patient example in Figure 4 were wrongly assigned to background air due to a metal wire cerclage in the sternum leading to a systematic underestimation of the PET signal. Another constraint in the MR-based AC of this specific patient example 2 is the missing reference in the Dixon-VIBE MR images due to the signal voids for the accurate registration of the bone atlas (wrong location of the spine). Therefore, patient example 2 was excluded from this evaluation.

Overall, *n* = 111 ^124^I-positive lesions were detected in the PET/CT data sets of all 35 patients and all 43 examinations. In PET/MR *n* = 132 ^124^I-positive lesions were detected, independent of the choice of MR-based AC (with or without bone atlas). Thus, twenty-one iodine-positive lesions were missed out in PET/CT compared to PET/MR. Image reading entailed identifying up to five lesions per patient with focally increased radiotracer uptake that were then further quantified. Thus, SUVs were measured in 98/111 PET/CT lesions and in 111/132 PET/MR lesions. For a valid comparison of congruent lesions in PET/CT and PET/MR, the mean ± standard deviation (SD), the range, and the median of the measured SUV_mean_ and SUV_max_ of 98 lesions detected both in PET/CT and in PET/MR with and without bone AC were analyzed (Table 3). The relative difference of each detected and congruent lesion (*n* = 98) in PET/CT vs. PET/MR with and without bone was calculated. The average relative difference in the median for the SUV_mean_ between PET/CT and PET/MR with bone was 6.3% and 13.3% for the SUV_max_. The average relative difference in the median for the SUV_mean_ between PET/MR without bone and PET/MR with bone was −1.1% and −1.0% for the SUV_max_.

Note that MR-based AC with bone resulted in slightly higher SUVs than the CT-based AC reference. MR-based AC with missing bone information tended to underestimate the PET signal compared to MR-based AC with bone atlas, however, with only very minor differences. The extreme high uptakes in SUV (>1000 in the SUV range) given in Table 3 are explainable due to the remaining thyroid parts, which were not fully removed in thyroidectomy. Thus, in these regions, the iodine uptake might be higher than in iodine-positive lesions or metastasis resulting in larger relative differences and standard deviations. Additionally, voxels with stochastic noise in the small VOIs may contribute to large, measured differences.

The Bland–Altman plots in Figure 5A,B show the relative differences between PET/CT and PET/MR with bone atlas in corresponding corrected PET data and calculated relative PET differences. Considering all 98 congruent lesions measured in both hybrid imaging modalities, the mean increase in SUV_mean_ is 36.2 ± 105.2% and the mean increase in SUV_max_ is 33.5 ± 84.8%. Note that for a better depiction, outliers are not shown in Figure 5A,B.

The Bland–Altman plots in Figure 6A,B show within the PET/MR reconstructions the relative differences between MR-based AC map with bone atlas (reference) and MR-based AC map without bone atlas in the measured SUV_mean_ and SUV_max_ of 111 of 132 iodine-positive detected lesions in PET/MR. Considering all 111 lesions, the mean in SUV_mean_ was decreased by −2.0 ± 7.0% with missing bone information. The mean in SUV_max_ was decreased by −1.5 ± 3.0% with missing bone information. Maximal relative differences were measurable in lesions close to the bone. Note that for a better depiction, outliers are not shown in Figure 6A,B.

Table 4 depicts SUV_mean_ and SUV_max_ in detected iodine-positive lesions sorted according to their location within the patient (lesions close to bone, in the lungs, lymph node lesions, and thyroid lesions) for PET/CT and PET/MR (MR-based AC with bone atlas). In eight detected lesions close to the base of the skull, sternum, clavicle, and cervical vertebral bone PET/MR resulted in less PET signal than measured in PET/CT (SUV_mean_ −23.4% and SUV_max_ −24.1%). Moreover, in lung metastasis, PET/MR showed decreased SUVs compared to the PET/CT reference (SUV_mean_ −15.5 % and SUV_max_ −15.4%). In thyroid lesions and lymph node lesions, PET/MR resulted in an increased PET signal compared to PET/CT (lymph node: SUV_mean_ 10% and SUV_max_ 15.6%, thyroid: SUV_mean_ 16.9% and SUV_max_ 23.9%).

One patient example is shown in Figure 7, who underwent a whole-body PET/CT examination 17 h and 50 min post-^124^I administration and a subsequent head-neck PET/MR 19 h and 16 min post-administration. Four iodine-positive and congruent lesions were detected in this patient in all three PET reconstructions. Due to the different patient positions within the PET/CT compared to PET/MR, lesion #2 could not be displayed in PET/CT in this example slice (Figure 7).

The relative differences in measured SUV_max_ values between PET/CT and PET/MR with bone AC were 27.2% in lesion #1, 143.1% in lesion #2, 13.2% in lesion #3, and 43.5% in lesion #4. The relative differences in measured SUV_max_ values between PET/MR with bone AC and PET/MR without bone AC were −3.5% in lesion #1, −5.4% in lesion #2, −11.3% in lesion #3, and −4.4% in lesion #4 (Figure 7).

## 4. Discussion

This retrospective single-administration, dual hybrid imaging comparison study including 35 patients with 43 examinations evaluated the intra-individual qualitative and quantitative differences of PET/CT and PET/MR of patients with DTC after thyroidectomy using the radiotracer ^124^I. Additionally, the isolated impact of improved MR-based AC on PET quantification using a bone atlas was analyzed in the PET/MRI data.

In this study, the overall number of thyroid lesions in the head-neck region detected by PET/MR with *n* = 132 was higher than the number of lesions detected with PET/CT, *n* = 111. Of these detected lesions, up to *n* = 5 lesions per patient and hybrid imaging modality were then further evaluated quantitatively. This resulted in a subset of *n* = 98/111 lesions in PET/CT and *n* = 111/132 in PET/MR that were further quantified. Of these, only the *n* = 98 congruent lesions, detected in both hybrid imaging modalities in the head-neck region were compared in an intra-individual comparison. In this comparison, the quantitative evaluation revealed higher mean and median values for SUV_max_ (36.3 ± 84.9%) and SUV_mean_ (36.2 ± 105.2%) for lesions measured in PET/MR when compared to lesions measured in PET/CT, respectively.

Despite these results in overall number and quantification differences, the results in general terms are in the range of—and comparable to—other studies comparing PET/CT vs. PET/MR in other body regions and using different radiotracers [24,25,26]. In this context, it has to be noted that this study specifically was set up as a retrospective, single-administration, dual hybrid imaging study where each hybrid imaging modality was independently used with its own and established clinical protocol and PET reconstruction parameters [24]. This was intended to identify commonalities in the lesion detection performance but also to quantify existent differences in a clinical hybrid imaging setting.

Consequently, beyond unifying the post-injection starting time of hybrid imaging acquisition for both exams, no further specific efforts were made to homogenize the PET reconstruction parameters or to cross-calibrate the PET/CT and PET/MR systems used in this study [27,28]. This implies that numerous methodological factors potentially contribute to the measured quantitative differences in overall detected lesions and in the intra-individual quantitative comparison of the detected congruent lesions. These factors are discussed in more detail in the following sections.

Numerous methodological aspects of our study may contribute to the resulting qualitative and quantitative differences in overall lesion number and measured SUV values in congruent lesions. Foremost, retrospective intra-individual comparison studies with two separate examinations on two fundamentally different hybrid imaging modalities PET/CT and PET/MR have resulted in differences within the quantitative range that were also found in the present study [24,26]. Two subsequent and independent hybrid examinations following a single administration of radiotracer lead to the fact that both examinations and data acquisitions start at different post-administration times. Depending on the tracer and examination protocol, this time difference may lead to differences in biodistribution of the tracer at the time of data acquisition. This may influence lesion conspicuity and/or activity quantification. In our study, however, this aspect of different post-administration starting times has been considered as far as practically possible in a clinical setting. The half-life time of ^124^I with 4 days, 4 h, and 13 minutes is rather long compared to the differences in post-administration starting times for both hybrid-imaging examinations. PET/CT exams on average started at 24:35 ± 2:47 hh:mm and PET/MR exams on average started at 28:53 ± 5:07 hh:mm post administration of the tracer. In all patients, the PET/CT exam was conducted first, followed by the PET/MR exam. Both factors, using a tracer with a rather long half-life time and applying a rather homogeneous protocol regarding post-administration starting times, reduced the potential quantitative effects of tracer dynamics on measured lesion activity.

Further methodological aspects with potential quantitative impact on PET measurements in this dual hybrid imaging study result from the fact that each patient was examined on two different hybrid PET systems in two independent exams. Here, fundamental differences between the PET detectors (hardware, geometry, electronics, time-resolution, sensitivity, etc.) and the PET acquisition parameters (e.g., 4 min acquisition per bed position and TOF detection for PET/CT vs. 20 min no-TOF for PET/MR) and reconstruction parameters (e.g., different OSEM reconstruction parameters and reconstructed spatial resolution, etc.) will inherently impact the measured results. No specific efforts were made to homogenize the PET reconstruction parameters or to cross-calibrate the PET/CT and PET/MR systems used in this study [24,27,28]. Instead, in this retrospective study, the PET reconstruction parameters for each hybrid imaging modality were kept according to its own established clinical protocol and PET reconstruction parameters [24]. Consequently, these aspects may quantitatively affect the SUV measurements in our study [27].

In addition to the PET acquisition and reconstruction parameters, the different attenuation correction methods in PET/CT and PET/MR account for differences in PET quantification. This aspect (AC in PET/CT vs. PET/MR) has been evaluated in more detail in this study as will be discussed in the following. It has to be noted, though, that only principal differences between CT-AC and MR-AC can be discussed in this context. The individual quantitative impact of each of the different AC methods on specific lesions cannot be evaluated independently of all the other parameters discussed earlier.

Fundamentally, different methods are used in PET/CT and PET/MR exams for attenuation and scatter correction and, moreover, patient positioning is also different. While in PET/CT, the patient is positioned with arms up, in PET/MR the patient is positioned with the arms resting along the body [4,5,24,26]. In a head-neck exam, this leads to the effect that a much larger portion of the photon-attenuating patient tissues (e.g., arms and shoulders) are located in the PET field of view during data acquisition in PET/CT compared to the PET/MR exam. This attenuation then needs to be corrected with the appropriate AC method. Further principal differences between CT-based AC and MR-based AC are that CT-based AC provides continuous LAC values for all tissues including bone [16]. MR-based AC on the other hand only provides four different tissue classes that are derived by image segmentation of MR images that do not directly represent a measure for photon attenuation [5,14,15]. Furthermore, bone tissue in MR-based AC is added as a separate tissue class with an atlas model providing LAC for major bones such as the skull, the spine, and the pelvic bones [17]. The MR-based bone atlas is an efficient and robust method to add attenuation information of bone tissue in the MR-based AC [17,19,20], but this model-based approach is only an approximation of the true bone anatomy and the position and LAC of bone may vary between the model and individual patient anatomy. Especially in patients with anatomical abnormalities (e.g., a very high/low BMI), the bone model might result in registration errors [29]. Investigating this specific aspect further in this study demonstrates that in the head-neck region the spine and skull in CT-based AC and in MR-based AC qualitatively show very comparable results (Figure 3). Also quantitatively, the intra-individual comparison of the bone volumes in the head-neck region revealed similar results for CT-based AC (4.48 ± 1.08%) and for MR-based AC (3.99 ± 0.96%) measured as the bone volume portion of the overall attenuating tissue volume (Table 2).

The different methods for tissue AC also revealed different artifacts in the head-neck region that may hamper diagnostic imaging in local regions, and, furthermore, have a potential impact on PET quantification as has been shown also in previous studies [14,23,30]. While CT-based AC in the head-neck region frequently shows streak artifacts due to dental implants and fillings, which may lead to locally increased LAC values in the CT-based AC maps associated with a local bias towards overcorrection of lesion activities [31], this was not observed in the MR-based AC data in this study (Figure 4). On the other hand, wire cerclages and other larger metal implants in the head-neck region may lead to local signal voids in the MR-based AC maps that then may lead to a local bias towards undercorrection of lesion activities (Figure 4). Nevertheless, the observed differences between CT-based AC and MR-based AC regarding bone volumes and observed artifacts in this study were considered minor and the overall quantitative impact of these two aspects on the study results is neglectable.

Beyond the qualitative and quantitative aspects of the PET/CT and PET/MR comparison discussed above, in this study, the isolated relative quantitative impact of adding bone as a further tissue class in MR-based AC was additionally investigated. The five-compartment MR-based AC employing the bone model served as a reference [17], while the previous standard MR-based AC using only four-compartments [14,15] was used for a second PET reconstruction of each patient data set. This allowed for measuring the isolated impact of both MR-based AC methods on each patient data set. As a result, missing bone information in the MR-based AC did not affect the overall clinical assessment of thyroid carcinoma in this ^124^I-PET/MR study. Overall, 132 lesions could be detected in both PET/MR reconstructions, with and without bone atlas in the MR-based AC. Comparing the improved five-compartment AC map (reference) with the previous standard four-compartment AC map in PET/MR, the overall difference in SUV_mean_ due to the addition of the bone model for the 111 of 132 congruent lesions that were further quantified was only small with −2.0% ± 7.0% (Figure 6). The relative quantitative impact of lesions located close to the bone was slightly higher than for soft tissue lesions located distant from the bone. This observation has also been reported by previous studies investigating the relative impact of bone in MR-based AC [16,17,19,20]. Nevertheless, in single lesions (close to the bone) relative differences in SUV_mean_ < −10% were calculated when neglecting bone AC (Figure 6). Thus, the individual impact of improved MR-based AC on each patient in the context of thyroid ^124^I PET/MR should be considered carefully.

Hybrid imaging with PET/CT and PET/MR using the radiotracer ^124^I in patients with differentiated thyroid carcinoma after thyroidectomy in this study has demonstrated robust and comparable diagnostic performance of both hybrid imaging modalities. The measured differences in lesion activity quantification of congruent lesions are in the range that was reported for previous single-administration dual hybrid imaging studies with PET/CT and PET/MR [11,32]. Such differences result from multiple methodological factors and challenges that are inherent to single-administration sequential PET studies on different PET systems [10] as discussed above. Our results support the examination of patients with differentiated thyroid carcinoma after thyroidectomy with any of the two-hybrid imaging modalities, ^124^I-PET/CT or ^124^I-PET/MR. The results of this study also imply that repeated exams of individual patients under therapy or for lesion dosimetry planning [6,27,33,34] should—whenever possible—be conducted with the same modality and, furthermore, on the same system to reduce methodological differences as far as possible [27]. Changes in individual lesion activity measured by PET ideally should be due to therapeutic effects only, and not due to changes in the hybrid imaging modality, methodology, imaging protocol, PET recon parameters, and/or AC method.

## 5. Conclusions

This retrospective single-administration dual hybrid imaging study evaluated the qualitative and quantitative differences between ^124^I-PET/CT and ^124^I-PET/MRI specifically in oncologic patients with differentiated thyroid carcinoma after thyroidectomy. The intra-individual comparison of 43 whole-body PET/CT and head-neck PET/MR patient examinations in this study demonstrated that the PET acquisitions in PET/MR have shown higher sensitivity when compared to PET/CT. The total number of ^124^I-avid lesions was higher for PET/MR when compared to PET/CT. The additional evaluation of PET/MR data corrected without and with bone atlas demonstrated that SUVs were slightly higher with the addition of bone information in the MR-based AC for all measured ^124^I-avid lesions.

## Figures and Tables

**Figure 1 cancers-14-03040-f001:**
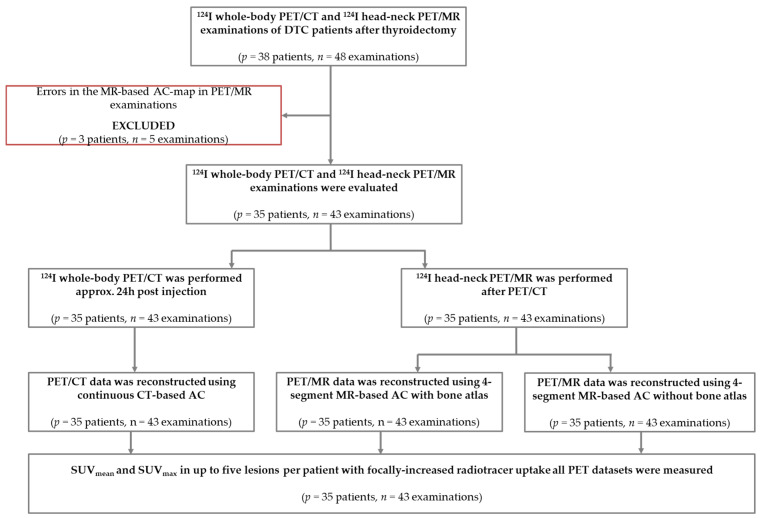
Flow chart of patients and PET/CT and PET/MR examinations analyzed in this study.

**Figure 2 cancers-14-03040-f002:**
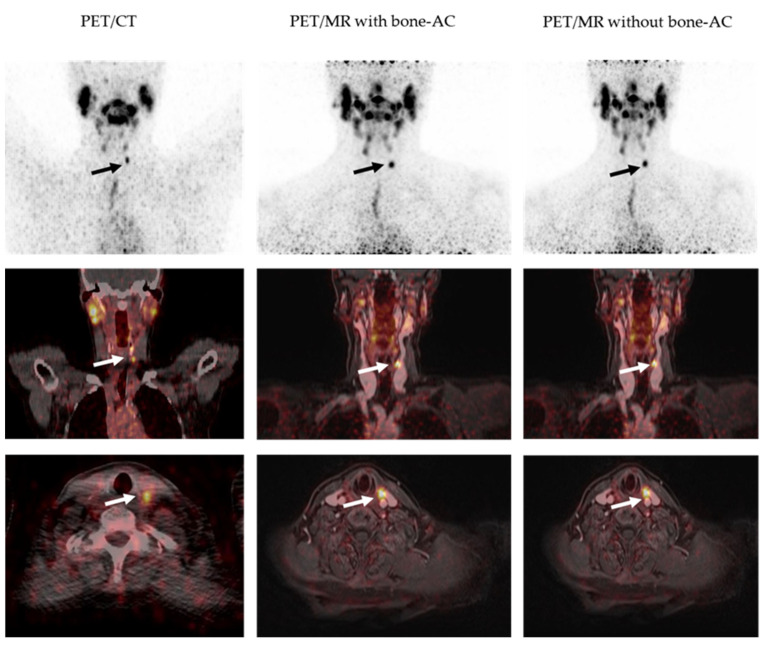
Intra-individual comparison of ^124^I-PET/CT and PET/MR examinations acquired approximately 24 h post-^124^I administration. From left to right column: PET/CT, PET/MR reconstructed with bone AC, and PET/MR reconstructed without bone AC. All hybrid data sets exemplarily show the uptake of a ^124^I-avid left cervical lesion (arrows). The lesion is well visible on all PET reconstructions (arrows) and presents with slightly improved visibility on the PET/MR-based reconstructions (middle and right column).

**Figure 3 cancers-14-03040-f003:**
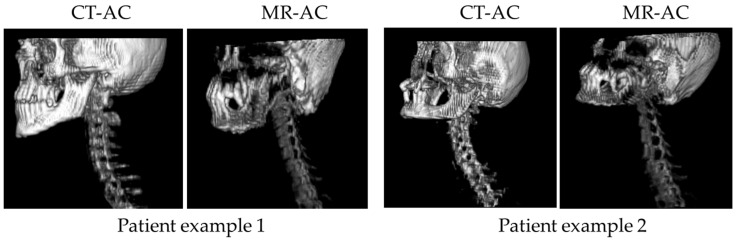
Comparison of 3D renderings of bone in CT-based and MR-based AC of two patient examples. While CT-based AC depicts the true bone anatomy (only skull and spine extracted from CT data for better comparability to MR-based AC) of each patient, bones in the MR-based AC result from an atlas-based approach and thus, represent the “best-match” between the bone model and the actual patient anatomy. Note that the spine bends differently in CT-based AC and MR-based AC due to the different patient positioning in PET/CT and PET/MR exams.

**Figure 4 cancers-14-03040-f004:**
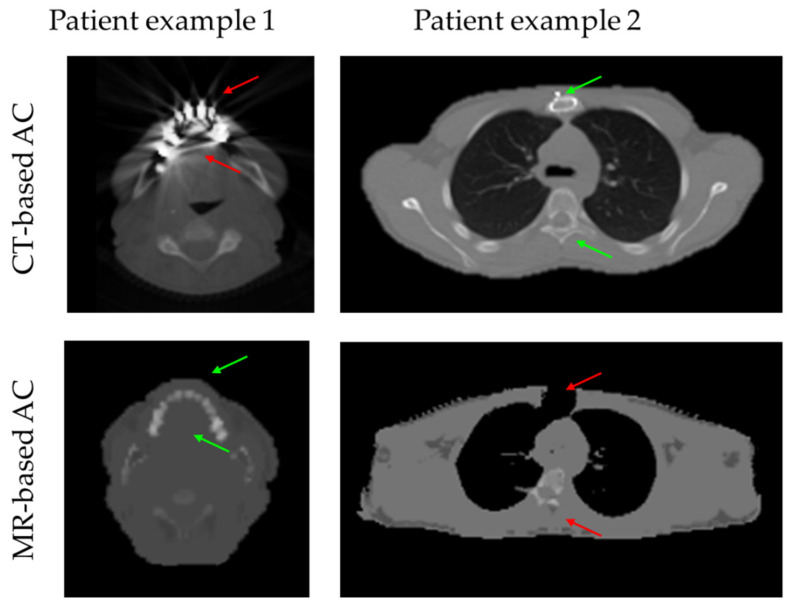
Two patient examples with artifacts in the AC maps based on CT or MR data are shown. The transaxial slice from patient example 1 (head) shows streak and beam hardening artifacts in the CT-based AC due to dental fillings and implants (red arrows). These CT-based AC artifacts may lead to systematic overestimation of PET signal in these regions after applying CT-based AC, while small dental fillings do not cause artifacts in MR-based AC (green arrows). Patient example 2 (thorax) shows MR-based AC artifacts due to metal implants in the sternum resulting in segmentation errors in the Dixon-VIBE AC map (upper red arrow), while the CT-based AC here shows the wire cerclage with its higher signal intensity (HU) but without noticeable artifacts (upper green arrow). Note the additional registration error of the bone atlas in patient example 2 due to segmentation errors in the MR-based AC itself (lower red arrow), while the CT-based AC shows the spine in the correct position (lower green arrow).

**Figure 5 cancers-14-03040-f005:**
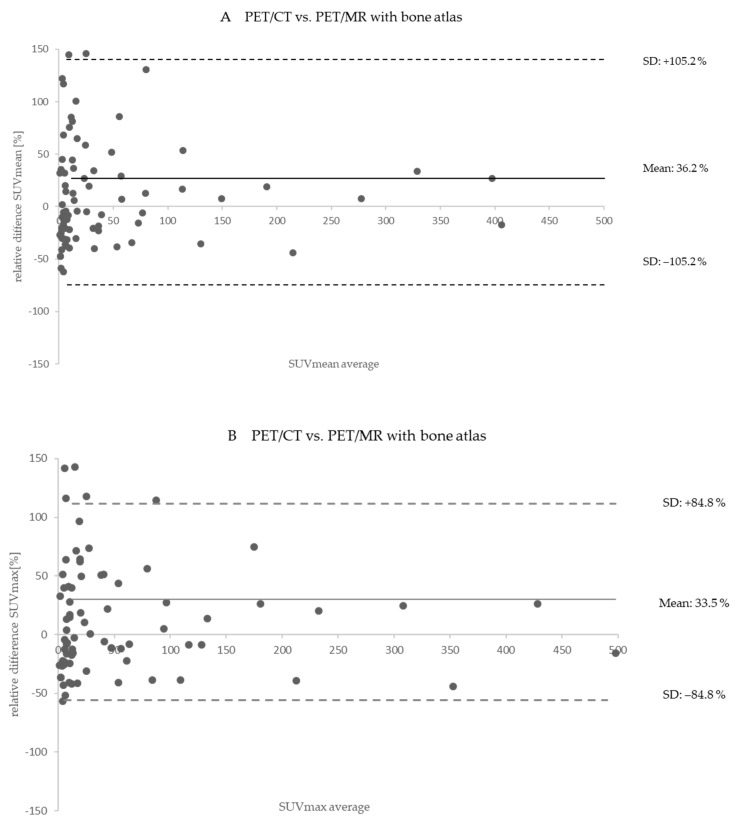
(**A**) Bland−Altman plot shows the relative quantitative differences in SUVmean for 98 congruent iodine−positive lesions measured in PET/CT (reference) vs. PET/MR corrected with MR−based AC with bone atlas. Note that PET/MR resulted in overall higher lesion SUV_max_ than the PET/CT measurements. (**B**) Bland–Altman plot shows the relative quantitative differences in SUV_max_ (**B**) for 98 congruent iodine-positive lesions measured in PET/CT (reference) vs. PET/MR corrected with MR-based AC with bone atlas. Note that PET/MR resulted in overall higher lesion SUV_max_ than the PET/CT measurements.

**Figure 6 cancers-14-03040-f006:**
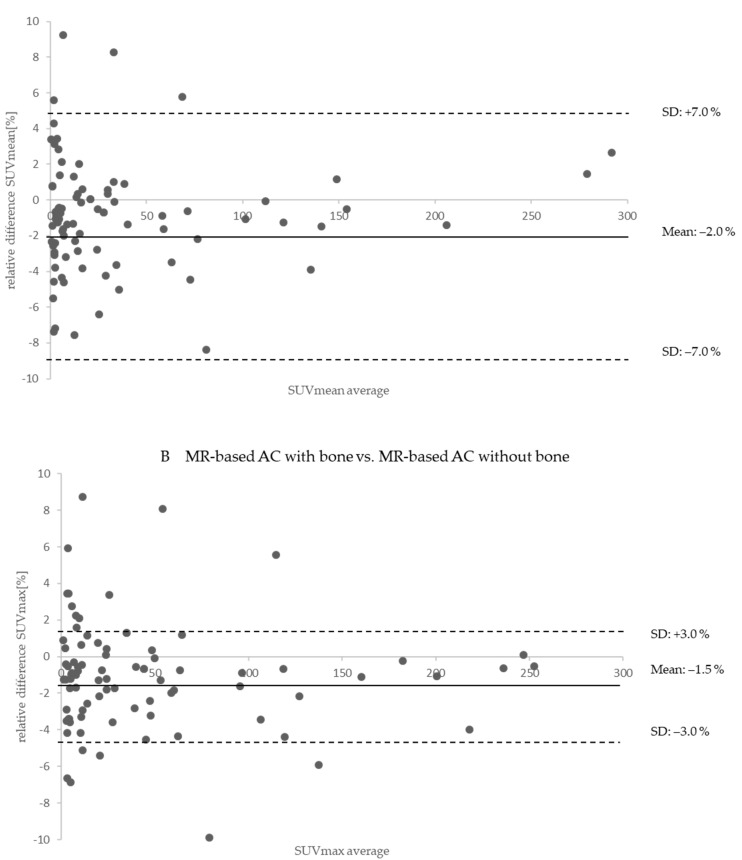
(**A**) Bland–Altman plot shows the relative quantitative impact on PET data in PET/MR between MR-based attenuation correction map (AC map) with bone atlas (reference) and MR-based AC map without bone atlas on standardized uptake values of SUV_mean_ in 111 detected iodine-positive lesions. MR-based AC with missing bone information tended to slightly underestimate the PET signal compared to MR-based AC with bone atlas (reference). (**B**) Bland–Altman plot shows the relative quantitative impact on PET data in PET/MR between MR-based attenuation correction map (AC map) with bone atlas (reference) and MR-based AC map without bone atlas on standardized uptake values of SUV_max_ in 111 detected iodine-positive lesions. MR-based AC with missing bone information tended to slightly underestimate the PET signal compared to MR-based AC with bone atlas (reference).

**Figure 7 cancers-14-03040-f007:**
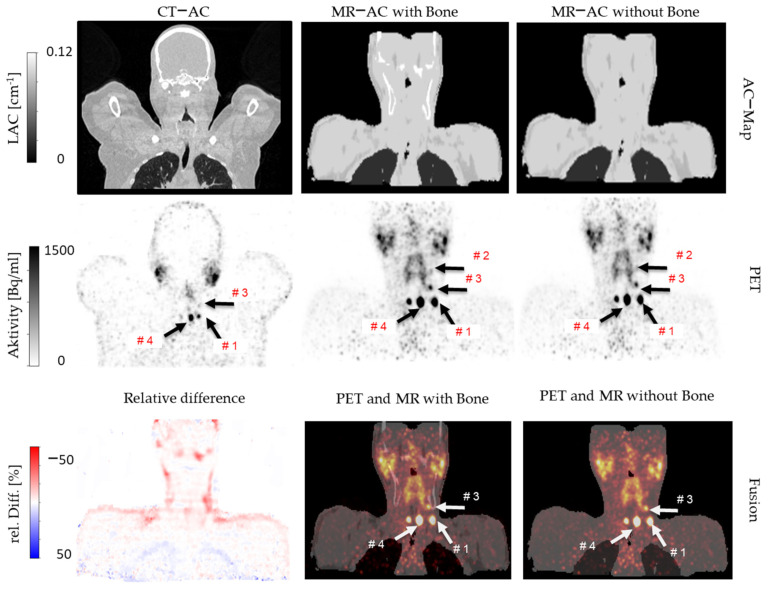
A patient example, who underwent a whole-body ^124^I−PET/CT and subsequent a head-neck ^124^I−PET/MR. Attenuation correction maps (AC maps, upper row), corresponding PET data (middle row), fusion images, and a relative difference image (lower row) are given. Exemplarily, three of five measured lesions are marked in the PET images (lesions #1, 3, 4 marked with white arrows, here visible in PET/CT and PET/MR data). Relative differences in PET signal between MR-based AC with vs. without bone show only small positive differences (red color) mostly in regions where the bone atlas added bone (spine and skull).

**Table 1 cancers-14-03040-t001:** Detailed patient and examination information. Values are given in mean ± standard deviation.

Patient No	Exam No	Sex	Age [Years]	BMI [kg/m^2^]	Tracer	Administered Activity [MBq]	Post Administration Time [hh:mm] PET/CT	Post Administration Time [hh:mm] PET/MR
35	43	23 f, 12 m	52 ± 18	27.2 ± 5.5	^124^I-NaI	34.2 ± 9.9	24:35 ± 2:47	28:53 ± 5:07

**Table 2 cancers-14-03040-t002:** Comparison of CT- and MR-based bone information in the attenuation correction maps (AC maps) of all patient data sets. Bone volume and corresponding linear attenuation coefficients (LAC) were measured in patients’ skull and spine (head-neck region). Note that the last column refers to the small volume and overall small percentage of voxels with high LAC in CT-based AC that is caused by dental artifacts.

CT- and MR-Based AC	Bone Volume * [%] Skull/Spine	LAC [cm^−1^] (Min–Max)	LAC [cm^−1^] (Mean ± SD)	LAC _max_ ** (>0.250 cm^−1^) [% vom V_tot_]
CT-based AC	4.483 ± 1.082	0.111–0.266	0.134 ± 0.021	mean: 0.023
MR-based AC	3.994 ± 0.961	0.101–0.247	0.128 ± 0.002	-

* Calculated bone volume with regard to total volume (V_tot_) of the MR-based AC; ** LAC_max_ in CT-based AC excluding metal implants.

**Table 3 cancers-14-03040-t003:** Standardized uptake values (SUV_mean_ and SUV_max_) in all measured *n* = 98 congruent lesions in PET/CT and PET/MR with and without bone atlas in the attenuation correction map (AC map). Relative differences between PET/CT (reference) and PET/MR with bone information, respectively, relative differences between PET/MR with bone (reference), and PET/MR without bone information are provided.

Meassured Values	Comparison between PET/CT vs. PET/MR with Bone AC			Comparison between PET/MR with vs. without Bone AC		
	PET/CT	PET/MR with Bone	Relative Difference [%]	PET/MR with Bone	PET/MR without Bone	Relative Difference [%]
Measured No. of lesions/No. of ^124^I-avid lesions	98/111	98/132	18.9	98/132	98/132	0
SUVmean mean ± SD (range) median	184.1 ± 472.3 (0.8–2669.8) 12.3	235.8 ± 575.0 (0.6–3036.7) 20.8	36.2 ± 105.2 (−67.9–720.4) 6.3	235.8 ± 575.0 (0.6–3036.7) 20.8	231.0 ± 564.6 (0.6–3065.3) 18.8	−2.2 ± 7.2 (−63.6–9.2) −1.1
SUVmax mean ± SD (range) median	296.8 ± 754.2 (1.5–4640.2) 20.6	393.6 ± 970.6 (1.1–5153.5) 34.6	36.3 ± 84.9 (−69.8–347.4) 13.3	393.6 ± 970.6 (1.1–5153.5) 34.6	386.3 ± 953.3 (1.1–5217.9) 35.0	−1.4 ± 3.4 (−11.9–8.7) −1.0

**Table 4 cancers-14-03040-t004:** Detected iodine-positive lesions and measured standardized uptake values (SUV_mean_ and SUV_max_) were sorted according to their location within the patient (lesions close to bone, in the lungs, lymph node lesions, and thyroid lesions) for PET/CT and PET/MR. Note that SUV measures in lesions close to bone or in the lungs are higher in PET/CT than PET/MR, while SUV measures in lymph node lesions or the thyroid are higher in PET/MR than PET/CT.

Detected Lesions Locations	PET/CT			PET/MR with Bone Atlas		
	No. of Lesions	Suv_mean_ Mean ± SD (Range) Median	SUV_max_ Mean ± SD (Range) Median	No. of Lesions	Suv_mean_ Mean ± SD (Range) Median	SUV_max_ Mean ± SD (Range) Median
Bone	8	4.73 ± 3.30 (1.94–11.85) 3.42	8.22 ± 5.90 (3.13–21.13) 6.12	8	3.95 ± 2.35 (1.55–7.37) 2.63	6.81 ± 3.84 (2.80–12.45) 4.91
Lung	8	68.49 ± 91.56 (4.49–275.12) 29.45	113.26 ± 150.85 (7.35–452.28) 48.12	8	44.30 ± 51.58 (4.24–154.46) 22.89	71.96 ± 83.58 (7.62–253.31) 36.98
Lymph nodes	46	61.97 ± 141.68 (0.81–775.43) 11.94	90.70 ± 192.81 (1.50–944.31) 18.10	46	97.76 ± 321.15 (0.59–2154.96) 22.69	165.02 ± 558.39 (1.11–3763.73) 37.23
Thyroid	36	405.64 ± 708.19 (0.97–2669.82) 30.64	665.13 ± 1131.25 (1.67–4640.19) 49.88	36	506.18 ± 806.35 (0.86–3036.71) 37.60	843.15 ± 1356.21 (1.60–5153.51) 63.79

## Data Availability

The data is available upon request from corresponding author.

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
