# Peer review of "Intra-Individual Comparison of 124I-PET/CT and 124I-PET/MR Hybrid Imaging of Patients with Resected Differentiated Thyroid Carcinoma: Aspects of Attenuation Correction"

_cancers, 2022, doi:10.3390/cancers14133040_

Round 1

Reviewer 1 Report

row 164: side lenght is in mm

row 255: you could emphasize that all the lesions dected by PET/CT were detected by PET/MR. I think it should be usefull to report how many lesions were dected by PET/CT out of the field of PET/MR.

Table 2: verify that ** is in the right position

I note a light contradiction in the affermation that PET/MR is only head and neck and reporeted data on lung lesions (all in apex?) and Fig. 3.

Correctly you say (rows 370-410) that quantitative data are not comparable between PET/CT and PET/MR but you report a comparison in conclusion. I think it should be better to limit conclusive considerations to say that PET/MR or, better, the PET acquisition in MR ,has shown higher sensitivity .

Author Response

Dear Editors,

We thank the editors for the time and effort invested in reviewing our work. The valuable comments have helped us very much in improving the manuscript. The table of the reviewer comments and our answers to each question have been added below. We hope that we could address all points appropriately in this revision.

Please see the attachment for point-by-point respose.

Kind regards from all authors

Reviewer 2 Report

All patients received a whole-body 99 (skull base to mid-thigh) PET/CT examination and an additional head- neck PET/MR al- 100 lowing for an intra-individual comparison. It would be useful to provide some images of the same modalities

Approximately 24 h post 124I administration (physical half-life: 4.2 days). Please provide some references regarding acquisition timing.

Author Response

Dear Editors,

We thank the editors for the time and effort invested in reviewing our work. The valuable comments have helped us very much in improving the manuscript. The table of the reviewer comments and our answers to each question have been added below. We hope that we could address all points appropriately in this revision.

please see the attachment for our point-by-point response.

Kind regards from all authors

Reviewer 3 Report

In this manuscript, the authors performed a retrospective study to evaluate the qualitative and quantitative differences of 124-iodine PET/CT and PET/MR of oncologic patients with differentiated thyroid carcinoma after thyroidectomy. Besides, The impact of improved MR-based attenuation correction (AC) using a bone atlas was also analyzed. Forty-three patients underwent 124I PET/CT and PET/MRI on the same day. In PET/CT, CT-based attenuation correction was used. As for PET/MRI, four-segment MRI-based AC with or without bone atlas was performed. In result, 111 124I-avid lesions were detected in PET/CT, while 132 lesions were detected in PET/MRI. The median in SUVmean for congruent lesions measured in PET/CT was 12.3. In PET/MR, the median in SUVmean was 16.6 with bone in MR-based AC. The authors concluded that (1)124I-PET/CT and 124I-PET/MR hybrid imaging of patients with DTC after thyroidectomy provides overall comparable quantitative results in a clinical setting despite different patient positioning and AC methods. (2) The overall number of detected 124I-avid lesions was higher for PET/MR compared to PET/CT. (3) The measured average SUVmean values for congruent lesions were higher for PET/MR.

General consideration and major problem:

1.     Significance of Content: The significance of this manuscript for clinicians and oncologists is low.

2.     Major weakness: The main aim of this study was to evaluate the qualitative and quantitative differences between 124-iodine PET/CT and PET/MR of oncologic patients with differentiated thyroid carcinoma. The authors had admitted that many confounding factors would influence the qualitative and quantitative results of their studies and they did not correct these factors. It is not surprising that the SUVmean and detected lesion number were higher in their PET/MRI scans because of longer scanning time per bed, arm-down position, etc. The SUV and detected lesion number could be similar if this is a prospective study and the major confounding factors are controlled.

Author Response

(The authors gave the same response as above.)
